# Unravelling the Diversity and Abundance of the Red Fox (*Vulpes vulpes*) Faecal Resistome and the Phenotypic Antibiotic Susceptibility of Indicator Bacteria

**DOI:** 10.3390/ani12192572

**Published:** 2022-09-26

**Authors:** Diana Dias, Dário Hipólito, Ana Figueiredo, Carlos Fonseca, Tânia Caetano, Sónia Mendo

**Affiliations:** 1CESAM and Department of Biology, Campus de Santiago, University of Aveiro, 3810-193 Aveiro, Portugal; 2Department of Biology, Faculty of Veterinary Medicine, University of Zagreb, Heinzelova 55, 10000 Zagreb, Croatia; 3Department of Bioscience & CEES, University of Oslo, Blindernvn, 31, 0371 Oslo, Norway; 4ForestWISE—Collaborative Laboratory for Integrated Forest & Fire Management, Quinta de Prados, 5001-801 Vila Real, Portugal

**Keywords:** antibiotic resistance genes (ARGs), *E. coli*, *Enterococcus* spp., mobile genetic elements (MGEs), qPCR, wildlife

## Abstract

**Simple Summary:**

Antimicrobial resistance was considered one of the major concerns of the twenty-first century by the World Health Organization in 2014. A holistic approach known as “One Health” recognizes the connections and interdependence between the health of people, domestic and wild animals, plants, and the ecosystem. The red fox is the most widespread wild canid in Europe that adapts easily and is distributed in natural environments and urban and peri-urban areas due to its increasing abundance. Foxes are reservoirs and disseminators of antibiotic resistance and zoonotic agents. They interact with watercourses, soils and livestock, and although they have no gastronomic interest, they are a game species, highlighting the potential risk of contamination between them and the hunters. Our main goal was to characterize antibiotic resistance in red foxes. Several clinically relevant antibiotic resistance genes were identified, as well as multidrug-resistant bacteria.

**Abstract:**

The WHO considers that antimicrobial resistance (AMR) is among the ten greatest global public health risks of the 21st century. The expansion of human populations and anthropogenically related activities, accompanied by the fragmentation of natural habitats, has resulted in increased human–wildlife interaction. Natural ecosystems are therefore subjected to anthropogenic inputs, which affect the resistome of wild animals. Thus, urgent multisectoral action is needed to achieve the Sustainable Development Goals following the One Health approach. The present work falls within the scope of this approach and aims to characterize the AMR of the faecal microbiome of the red fox (*Vulpes vulpes*), an opportunistic and generalist synanthropic species whose abundance has been increasing in urban and peri-urban areas. A high number of antibiotic resistance genes (ARGs) and mobile genetic elements (MGEs) were screened and quantified using a high-throughput qPCR approach, and the antimicrobial susceptibility of cultivable *E. coli* and *Enterococcus* spp. were assessed interpreted with both ECOFFs and clinical breakpoints. The most abundant ARGs detected confer resistance to trimethoprim and tetracyclines, although the first were absent in one of the locations studied. Several ARGs considered to be threats to human health were identified in high relative abundances (*bla*_TEM_, *ermB*, *aadA*, *tetM*, *tetW*, *tetL*, *drfA1* and *drfA17*), especially in the geographical area with greater anthropogenic influence. Although at a low percentage, resistant and multidrug-resistant (MDR) *E. coli* and *Enterococcus* spp. were isolated, including one MDR *E. coli* showing resistance to 12 antimicrobials from 6 different classes.

## 1. Introduction

Antimicrobial resistance (AMR) is a global health crisis, considered by many to be a silent pandemic that jeopardizes a century of medical progress and threatens the achievement of the Sustainable Development Goals [1,2,3]. Environmental microorganisms are the largest natural producers of antibiotics. Thus, antibiotic resistance exists and has evolved long before the discovery and development of antibiotics to fight infectious diseases. However, the emergence of clinical resistance has been accelerated by selective pressure of genes often mobilized from the environmental resistome [4]. Besides the classic mechanisms of horizontal gene transfer (HGT), mobile genetic elements (MGEs) such as genomic islands, insertion sequences (IS), transposons and integrons, among others, might facilitate the spread of such bacterial resistance traits [5]. As a result, many commensal bacteria in the human gut microbiome that were once considered harmless are now considered harmful pathogens after the acquisition and activation of such traits [6]. Furthermore, the use of structurally similar compounds and/or belonging to the same class of drugs in food-producing animals and in humans seem to have exacerbated the emergence of resistance in bacteria relevant to public health [7]. As in humans, 30–90% of the antibiotics used in animals are released into the environment through urine and faeces, and most of the animal waste is used as manure in agriculture. Therefore, they are considered the main source of antibiotic-resistant bacteria and antibiotic resistance genes (ARGs) dissemination in the environment [8,9]. The interaction and contact of wild animals with human activities seem to influence their antibiotic resistance profiles [10,11] and may therefore contribute to the AMR dissemination cycle. AMR surveillance systems should take a worldwide “One Health” strategy, which incorporates interdisciplinary and collaborative efforts to achieve optimal health for humans, animals, and the environment [7]. There are still few studies of AMR in wildlife when compared to those performed in humans and food-producing animals [11,12]. However, these studies are essential to better understand the dynamics of AMR in natural ecosystems. Likewise, the identification of sentinel wildlife species to monitor environmental antibiotic resistance is also needed [11,13].

The red fox (*Vulpes vulpes*) is the most widespread wild canid in Europe [14]. This species is a generalist and opportunistic carnivore, feeding on a wide variety of foods, including mice, rabbits, birds, small mustelids, eggs, fruits, seeds, and human waste [15]. The increasing abundance of foxes in urban and peri-urban areas, together with their behaviour, places this species as an epidemiological link between humans, livestock and natural environments [14]. Studies on red foxes have revealed that they are not only reservoirs of various parasites with zoonotic potential and veterinary relevance [16], but are also a source of AMR bacteria [17,18,19,20,21], which has led them being proposed as good environmental AMR sentinels [19,21]. AMR research in these animals has been mainly focused at characterizing the antimicrobial susceptibility of cultivable bacteria (e.g., *E. coli*, *Enterococcus* spp., *Salmonella* spp., and methicillin resistant *Staphylococcus aureus*) [17,18,22,23]. This approach has been the basis of AMR monitoring around the world, but different factors are measured using cultivable or non-cultivable methods [11,24] that allow for comparison with previous studies, while providing data for future research based on high-throughput metagenomic methods. As such, our goal was to characterize and quantify AMR in red foxes using a metagenomic approach and to assess the resistance profiles of *E. coli* and *Enterococcus* spp. isolates. The metagenomic approach applied was “ARG array 2.0” that in recent years, has facilitated the resistome characterization of a large variety of environmental samples [25]. The array is adaptable to the goals of each study and can be selected by the user, as it offers a high number of validated primer sets targeting a wide range of ARGs from the main antibiotic classes and MGEs [25]. The phenotypic approach was applied using two species that represent the most investigated groups on AMR surveillance of foodborne bacteria, which are used to evaluate water quality, may cause disease, and are common carriers of acquired ARGs that may be transmitted to human pathogens [26,27]. The AST results were interpreted according to clinical and epidemiological cut-offs (ECOFFs) since the latter are more appropriate for environmental studies. To the best of our knowledge, this is the first study to investigate the diversity and abundance of ARGs in red fox populations, combined with phenotypic data.

## 2. Materials and Methods

### 2.1. Study Areas and Sampling

From November 2017 to November 2019, 37 red fox faecal samples were collected (noninvasive sampling) from two different geographic locations (Appendix A): (i) Montesinho Natural Park (MontesinhoNP; *n* = 11), and (ii) Freita, Arada and Montemuro Mountains (FreitaAMM; *n* = 26). MontesinhoNP, with about 75.000 ha, located in the northeast of Portugal, bordering Spain, and is part of the Natura 2000 Network (site codes PTZPE0003 and PTCON0002). It is mostly a rural area, composed of small villages with low human and livestock densities and includes a National Hunting area. FreitaAMM, with about 70.000 ha, is located in the central-north of Portugal, and integrates the Natura 2000 Network (site codes PTCON0047 and PTCON0025). The municipalities that comprise FreitaAMM have a medium to high population density, although on the mountains, the population is dispersed through the valleys in small villages, and subsists on agriculture and pastoralism, from raising livestock, small ruminants, and bovines to producing milk and meat. This area is completely covered by different hunting grounds. A scientific study estimating the fox population size in the two locations is lacking. The faecal samples were collected from natural environments when considered fresh by experienced personnel and refrigerated at 4 °C, up to 24 h. One gram was smashed and diluted in 10 mL of Buffered Peptone Water (Liofilchem, Italy) and incubated at 37 °C, overnight, with aeration (180 rpm), to subsequently isolate *E. coli* and *Enterococcus* spp. strains. Five grams of each sample was transferred to a vial and stored at −80 °C for further DNA extraction. 

### 2.2. DNA Extraction and Pooling

Total DNA was extracted from each faecal sample (*n* = 23) with the DNeasy^®^ PowerSoil^®^ Kit (Quiagen, Germany), and processing of samples was performed as described by Dias et al. [28]. When using sensitive technology such as qPCR, sample pooling has proven to be very efficient [29]. Up to 8 extractions were used to obtain each DNA pool, which was analysed containing an equal concentration of DNA from each sample and was prepared and used for the high-throughput qPCR pre-screening analysis. Final high-throughput qPCR analysis was performed with 3 DNA pools: (i) pool 1 (*n* = 8) and pool 2 (*n* = 8) included DNA from FreitaAMM samples, and (ii) pool 3 (*n* = 7) was composed of DNA from MontesinhoNP samples. In each pool, the final DNA concentration was 20 ng/μL.

### 2.3. High-Throughput qPCR

The ARG qPCR array 2.0 [25] was used to identify and quantify ARGs and MGEs by Resistomap (Finland) and the array constitution and analysis were made in agreement with Dias et al. [28]. As mentioned above, a pre-screening was performed in a DNA pool template representing 23 red fox samples and using 384 primer sets that identify: (i) ARGs from the main antibiotic groups (aminoglycosides, β-lactams, macrolide-lincosamide-streptogramin B (MLSB), multidrug efflux-pumps, phenicols, quinolones, sulfonamides, tetracyclines, trimethoprim and vancomycin), (ii) genes that confer resistance to other antimicrobials such as nisin, bacitracin and antiseptics, and (iii) genes associated with MGEs and integrons. The results obtained were used to set up a customized array of 123 primer sets (97 primer sets for 91 different ARGs, and 26 primer sets to 23 different MGEs) and the 16S rRNA gene (Appendix A), which allowed for the quantification of ARGs and MGEs using the pools of DNAs from FreitaAMM (2 pools) and MontesinhoNP (1 pool), prepared as described above.

### 2.4. Isolation and Selection of E. coli and Enterococcus spp.

*E. coli* and *Enterococcus* spp. were isolated from the initial cultures of the faecal samples following the protocol described in [28]. Briefly, dilutions from each culture were seeded on selective culture media: MacConkey agar (Liofilchem, Italy) for *E. coli* and Slanetz and Bartley agar + TTC (Liofilchem, Italy) for *Enterococcus* spp., and the identity of the isolates was confirmed by colony-PCR using species-specific genetic markers, as described by Dias et al. [28]. One random colony of *E. coli* and *Enterococcus* spp. from each faecal sample was further subjected to AST.

### 2.5. Antibiotic Susceptibility Testing (AST)

AST was carried out by disk diffusion susceptibility testing and, according to the classes, the antibiotics tested were (μg/disc): (i) β-lactams: ampicillin (10), amoxicillin-clavulanic acid (30), cefoxitin (30), cefotaxime (5), ceftazidime (10), aztreonam (30), imipenem (10), (ii) aminoglycosides: amikacin (30), gentamicin (10), streptomycin (10) and tobramycin (10), (ii) quinolones: ciprofloxacin (5) and nalidixic acid (30), (iii) chloramphenicol (30), (iv) macrolide: erythromycin (15), (v) tetracycline (30), (vi) glycopeptides: teicoplanin (30) and vancomycin (5), (vii) streptogramin: quinupristin–dalfopristin (15), (viii) glycylcycline: tigecycline (15) and others: trimethoprim–sulfamethoxazole (25) and nitrofurantoin (100). AST was performed according to the European Committee on Antimicrobial Susceptibility Testing (EUCAST) guidelines, and *E. coli* ATCC 25922 and *E. faecalis* ATCC 29212 were used as quality controls. The interpretation of inhibition zone diameters (IZDs) was performed with clinical breakpoints and epidemiological cut-off values (ECOFFs). Clinical breakpoints defined by EUCAST were used to classify the susceptibility of strains, except for streptomycin, nalidixic acid, and tetracycline for *E. coli* and ampicillin, chloramphenicol, erythromycin, tetracycline, and gentamicin for *Enterococcus* spp., where CLSI breakpoints were applied [28]. The *E. faecalis* IZDs to quinupristin–dalfopristin (QDA) were not considered for the AST interpretation with clinical breakpoints, since *E. faecalis* is intrinsically resistant to this antibiotic [30,31]. The ECOFFs for *E. coli* and *Enterococcus* spp. were determined using the normalized resistance interpretation (NRI) technique [32] by testing bacteria isolated from the faeces of wild mammals collected by our group since 2017 (271 *E. coli* and 244 *Enterococcus* spp.) [28]. Strains were classified as multidrug-resistant (MDR) following the CDC definition of isolates resistant to, at least, one agent in three or more antibiotic classes.

### 2.6. Data Analysis

The NRI method was used with permission from the patent holder, Bioscand AB, TÄBY (Sweden), under the European patent No. 1,383,913, United States Patent No. 7,465,559. The automatic and manual excel programmes to determine NRI were made available by courtesy of Dr. P. Smith, Dr. W. Finnegan, and Dr. G. Kronvall. Venn diagram was obtained with the website tool http://bioinformatics.psb.ugent.be/webtools/Venn/ (accessed on 6 January 2022). The average values of the relative abundances of ARGs, the analysis of ARGs diversity, the log_10_ transformation of relative gene abundances, and the prevalence of phenotypic antimicrobial resistance were calculated with Microsoft Office Excel 2021. The plots of relative abundance of high-threat ARGs and the box plot showing the average and standard deviations of the relative gene abundances were constructed using the Plotly Chart Studio [33].

## 3. Results and Discussion

### 3.1. Overview of ARGs and MGEs Found in Red Fox

Some genotypic studies have investigated the occurrence of ARGs on cultivable bacteria recovered from red fox [21], mainly those encoding extended-spectrum β-lactamases and/or AmpCs [19,34]. However, studies on its resistome with metagenomic approaches are scarce [35]. Thus, herein, a high-throughput qPCR method was employed, using faecal DNA to gain insights into the diversity and abundance of the faecal resistome of red fox. After the first screening (384 assays), 123 assays that amplify 91 different ARGs were selected, conferring resistance to 10 groups of antibiotics, and 23 different MGEs to analyse samples collected in two different geographic locations (FreitaAMM and MontesinhoNP). The results obtained are comparable to those of other studies employing the same method, but where other types of samples were analysed, namely, pig faeces (108 ARGs from 11 groups and 28 MGEs) [36], wild boar faeces (62 ARGs from 9 groups and 20 MGEs) [28], red deer faeces (41 ARGs from 7 groups and 14 MGEs) [37] and soils receiving swine and dairy manures (77 ARGs from 8 groups and 12 MGEs) [38]. 

In general, the ARGs identified in red fox confer resistance mainly to aminoglycosides (24%), tetracyclines (20%), MLSBs (19%) and β-lactams (12%) (Figure 1A, Appendix A). The resistance mechanisms encoded by these ARGs include antibiotic deactivation (42%), cellular protection (33%), efflux pumps (21%) and other/unknown (4%) (Figure 1B). Regarding the MGE groups, genes encoding 10 IS, 6 transposases, 2 integrases, and 8 plasmid-associated genes were detected (Figure 1C).

According to the location, red foxes from FreitaAMM had a high diversity of ARGs and MGEs with 115 positive assays, whereas foxes from MontesinhoNP tested positive in 62 assays (Figure 2A). DNA from FreitaAMM samples amplified a high number of unique genes (*n* = 61) when compared to MontesinhoNP (*n* = 8), and 54 genes were found in the samples of both study areas. The 90 ARGs identified in foxes living in FreitaAMM belong mainly to aminoglycosides (26%), MLSBs (20%) and tetracyclines (18%) (Appendix A). In samples from MontesinhoNP, the 45 ARGs are mainly associated with the resistance to aminoglycosides (24%), tetracyclines (22%), and MLSBs (18%) (Appendix A). None of the tested ARGs of the trimethoprim group were detected in samples from MontesinhoNP (Figure 2C).

### 3.2. Abundance of AMR Genetic Determinants in Red Fox

The quantification of ARGs was made in relation to the abundance of the 16S rRNA gene in each sample. Their relative abundance varied between ca. 10^−6^–10^−2^, having an average of 1.97 × 10^−3^ (Appendix A and Figure 3). 

The abundance of ARGs found here for red fox was slightly higher than the range reported for wild boar faeces (which was between ca. 10^−6^–10^−2^, with an average of 6 × 10^−4^) [28], red deer faeces (which was between ca. 10^−6^–10^−3^, with an average of 9.85 × 10^−5^) [37], and soil ecosystems (10^−6^ to 10^−4^ copies/16S rRNA gene copy) [39], but lower than that found for livestock manure in almost 100 countries, which varied between 10^−3^ and 10^−1^/16S ribosomal RNA [40]. 

In general, the most abundant ARGs are linked to the resistance to trimethoprim (mean ca. 3.2 × 10^−3^), followed by tetracyclines (mean ca. 2.7 × 10^−3^), aminoglycosides (mean ca. 2.1 × 10^−3^) and β-lactams (mean ca. 2.1 × 10^−3^) (Figure 3, Appendix A). However, the three most abundant ARGs confer resistance to β-lactams (*bla_TEM_*; 2.9 × 10^−2^), macrolides (*ermB*; 2.01 × 10^−2^) and aminoglycosides (*aadA2*; 1.84 × 10^−2^) (Appendix A). These results were different from those reported in a study carried out in Poland, based on shotgun metagenomic sequencing, which identified tetracycline as the most abundant encoded resistance (being *tetQ* the most abundant ARG), followed by resistance to macrolides, β-lactams, and aminoglycosides [35]. In Central Chile, a study using faecal swabs from Andean foxes (*Lycalopex culpaeus*) also showed that the *tetQ* gene was among the most abundant ARGs, along with *tetW* [41]. Regarding MGEs, their relative gene abundances varied between ca. 10^−5^–10^−2^, with a mean of 4.5 × 10^−3^ (Appendix A and Figure 3).

The cumulative abundances (sums) for each class of ARG differed between the study areas (Figure 2D): (i) in FreitaAMM, similar cumulative abundances were observed for ARGs conferring resistance to aminoglycosides (29%), tetracyclines (22%) and MLSBs (19%), and (ii) in MontesinhoNP, the most abundant class was tetracycline, making over 71% of the sum of the total ARGs abundances. The most abundant ARGs classes also varied according to the geographical origin of samples. In FreitaAMM, ARGs encoding resistance to trimethoprim were highly abundant (mean ca. 3.2 × 10^−3^), whereas no ARG of this class was detected in MontesinhoNP. Following trimethoprim, β-lactam (mean ca. 2.9 × 10^−3^), aminoglycoside (mean ca. 2.8 × 10^−3^) and tetracycline (mean ca. 2.6 × 10^−3^) ARGs followed (Figure 2B, Appendix A). In MontesinhoNP, the most abundant ARGs are involved in the resistance to tetracyclines (mean ca. 2.9 × 10^−3^), MLSBs (mean ca. 5.8 × 10^−4^), vancomycin (mean ca. 4.9 × 10^−4^) and β-lactams (mean ca. 4.4 × 10^−4^) (Figure 2B, Appendix A).

Regarding MGEs, *Tnp*A transposase was the one detected in greater abundance (6.43 × 10^−2^), followed by the insertion sequence *ISE*fm1 (3.79 × 10^−2^). Class 1 integrase gene (*intI1*; 2.62 × 10^−2^), considered to be an anthropogenic bioindicator, was more abundant than class 3 (*intI3*) (4.87 × 10^−4^) (Appendix A).

### 3.3. Environmental Indicators and ARGs Associated with Human Health

Berendonk et al. (2015) [27] and Gillings et al. (2015) [42] suggested the use of key indicator genes to assess the antibiotic resistance status in the environment. Nine of these genes were detected in the present study (*intI1*, *sul2*, *bla*_TEM_, *ermB*, *ermF*, *tetM* and *aph* on both locations; *vanA* only in MontesinhoNP; and *bla*_CTX-M_ only in FreitaAMM), suggesting that the fox populations in this study are subjected to the input of anthropogenic ARGs. Recently, to facilitate the interpretation of risks to human health, Zhang et al. (2021) [43] proposed ranks for ARGs. Those considered “current threats” were included in Rank I and “future threats” in Rank II. Twenty-seven ARGs belonging to these two ranks were identified on the red fox resistome: 12 ARGs in the MontesinhoNP population, and twice as much (24 ARGs) in red foxes from FreitaAMM (Figure 4). One-third of these ARGs (*bla*_TEM_, *ermB*, *aadA*, *tetM*, *tetW*, *tetL*, *drfA1* and *drfA17*) were observed at high relative abundances (10^−3^–10^−2^), especially in samples collected from FreitaAMM (Figure 4). Although ARGs are found ubiquitously among human gut commensal species, high-risk ARGs were discovered to be enriched in pathogenic strains [43]. Thus, the higher occurrence and abundance of Rank I and II ARGs in FreitaAMM samples indicate that the animals of this region can be exposed to higher selective pressures, which is in accordance with the nature of this ecosystem, considered to have medium to high human density and medium livestock densities when compared to MontesinhoNP, which is less subjected to anthropogenic disturbances. 

Compared with the data available for red deer and wild boar inhabiting MontesinhoNP [28,37], it was concluded that red foxes have the highest diversity and abundance values of RankI and RankII ARGs, the red deer has the lowest, and the wild boar is in between. This corresponds to their proximity to humans (red fox > wild boar > red deer). Together, our results suggest that red foxes can acquire ARGs and MGEs through human sources and are possibly an important link in the chain of transmission from humans to other wild animals and the environment, being a good sentinel species for monitoring antimicrobial resistance in the environment, as suggested by Mo et al. (2018) [19].

### 3.4. Prevalence of Antibiotic Resistance in E. coli

It was possible to recover *E. coli* isolates from 86% of the faeces (*n* = 32), which is a value similar to that of a study carried out on foxes in Norway (82%) [19], but twice as much as reported in a study on foxes in Northern Portugal (42%) [21]. The resistance phenotype was analysed according to clinical breakpoints and to the ECOFFs, as the interpretation criteria used have a great influence on the resistance rates [44]. According to the clinical breakpoints, 34% (*n* = 11) of the strains were resistant to at least one of the antibiotics tested. The observed resistance was mainly for aminoglycosides and β-lactams (Figure 5A, Appendix A), whereas other studies conducted in Portugal with red foxes reported higher resistance of *E. coli* to streptomycin and tetracycline [21]. In FreitaAMM, 35% (*n* = 9) of the *E. coli* were resistant to at least one antibiotic, mostly to β-lactams and aminoglycosides. Of the *E. coli* isolated from MontesinhoNP, two isolates (18%) showed resistance to aminoglycosides and tetracycline. A MDR phenotype was detected for five strains (16%), all of them isolated from FreitaAMM (Appendix A), with one strain being resistant to 12 antibiotics from 6 classes. MDR *E. coli* were also isolated from foxes in central Italy, where two out of six strains exhibited that phenotype [20].

According to the ECOFFs calculated with the NRI [28], 50% (*n* = 16) of the *E. coli* were NWT for at least one of the antibiotics tested and mainly to quinolones, β-lactams, and tetracycline (Figure 5B, Appendix A). This is a higher rate than observed in red foxes from Norway (8%), where *E. coli* were mostly resistant to sulfamethoxazole, ampicillin, and tetracycline [19]. For instance, in the last 12 years in Portugal, the consumption of antibacterials for systemic use in the community and hospital sectors has been higher than in Norway [45]. Furthermore, in the last 2 years, the sale of antimicrobials for veterinary use in Portugal was 179.1 tonnes, which contrasts with the much lower value of 5.1 tonnes in Norway [46].

Even so, although high rates of resistance to some antibiotics were detected, these are less worrisome than those reported in a study conducted in Portugal about 10 years ago [21]. 

Considering each location, 42% of the strains (*n* = 11) with an origin in FreitaAMM had an NWT phenotype, mostly for quinolones and β-lactams. For MontesinhoNP, an NWT phenotype was identified in 45% (*n* = 5) of the isolates, mostly for the quinolone ciprofloxacin. The use of ECOFFs showed that resistance to ciprofloxacin, a commonly used quinolone and very stable ex vivo antibiotic [47], might be rising.

### 3.5. Prevalence of Antibiotic Resistance in Enterococcus spp.

*Enterococcus* spp. isolates were recovered from 89% of the faecal samples (*n* = 33) and belonged to the following species: *E. faecalis* (49%), *E. faecium* (39%) and *E. hirae* (12%). Their prevalence was slightly lower in our study (89%) than that reported in the study conducted on foxes in Northern Portugal (96%) [21]. According to the clinical breakpoints, the AST showed that 73% (*n* = 24) of *Enterococcus* spp. strains were resistant to at least one of the antibiotics tested, which is lower than the reported in a previous study with foxes from the North of Portugal [21]. In our study, higher percentages of resistance were observed for QDA, tigecycline, tetracycline and erythromycin (Figure 5C; Appendix A). Although in a different order, the highest resistance found by [21] was for the same antibiotics: tetracycline > erythromycin > QDA (tigecycline is not considered, as it was not assessed by [21]). Resistance to glycopeptides, important for the treatment of severe human infections, was identified only for 6% of the strains to teicoplanin, whereas no resistance was detected for vancomycin. By location, 71% (*n* = 17) of the isolates from FreitaAMM showed resistance to at least one antibiotic, and especially for streptogramin, tetracycline, glycylcycline and macrolide classes. Regarding the strains from MontesinhoNP, 78% (*n* = 7) were resistant to at least one antibiotic, mainly to streptogramin and glycylcycline classes. A low prevalence of MDR enterococci was identified (3%; one strain from MontesinhoNP; Appendix A), contrary to the high rate reported in a study carried out with wild Pampas foxes (*Lycalopex gymnocercus*) in the Brazilian Pampa biome (63%) [48]. The authors suggest that the high MDR frequency was associated with the proximity to anthropogenic activities, since foxes are indifferent to the presence of humans and often share the same habitat [48].

According to the ECOFFs calculated with the NRI method [28], 55% (*n* = 18) of the strains were NWT to at least one of the antibiotics tested, mainly for the aminoglycoside and macrolide classes (Figure 5D) (Appendix A). Considering each collection site, an NWT profile was detected for 54% (*n* = 13) of the strains from FreitaAMM, and mostly to aminoglycosides, tetracycline and macrolide classes. An NWT phenotype was observed in 56% (*n* = 9) of the strains from MontesinhoNP, mainly to macrolide, aminoglycoside and glycylcycline classes.

## 4. Conclusions

This study fits in the One Health concept, which raises awareness of potential reservoirs and sources of AMR in the environment. It was found that, in general, red foxes in Portugal have higher antimicrobial resistance rates than red foxes from Norway [19], which may be associated with the increased use and exposure to antimicrobials in both humans and domestic animals, and the proximity of red foxes to these populations. AMR studies with red foxes have reported higher resistance rates to tetracyclines, SXT and aminoglycosides, which were also observed in our study. A high diversity and number of ARGs was also identified, and to the best of our knowledge, this is the first study combining genotypical and phenotypical approaches to assess AMR in red foxes. The most abundant ARGs found in the faecal resistome of red foxes confers resistance to trimethoprim and tetracycline. ARGs considered threats to human health (Rank I and Rank II) were detected, one third of them with high relative abundances. The diversity and abundance of these ARGs were undoubtedly greater in animals living in the area with more human impact (FreitaAMM). In the future, the use of a reduced ARGs array targeting only highly relevant ARGs will allow for analysis of, affordably, a larger number of samples, to avoid the pooling, which is acceptable but might be a limitation in our study. Further studies are needed to better understand the dynamics and networks of AMR dissemination between the domains of the One Health approach. Finally, the number of MDR strains identified was relatively low, and ARGs normally associated with a high level of antibiotic resistance such as *bla_KPC_*, *bla_NDM_* and *bla_VIM_* were not identified, nor were *mcr*-1 and *mcr*-2, which confer resistance to the last-resort antibiotic, colistin.

## Figures and Tables

**Figure 1 animals-12-02572-f001:**
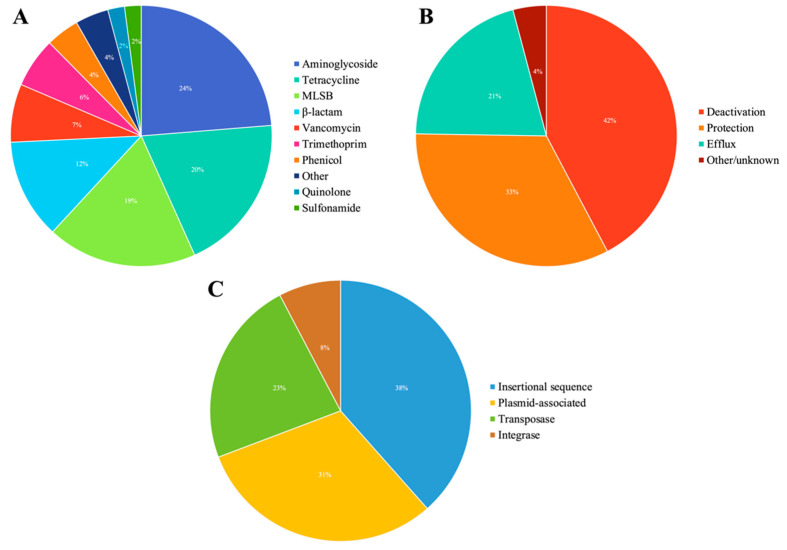
Diversity of ARGs identified in red fox faecal samples classified according to the antibiotic class to which they confer resistance (**A**), their resistance mechanism (**B**) and the types of MGEs (**C**).

**Figure 2 animals-12-02572-f002:**
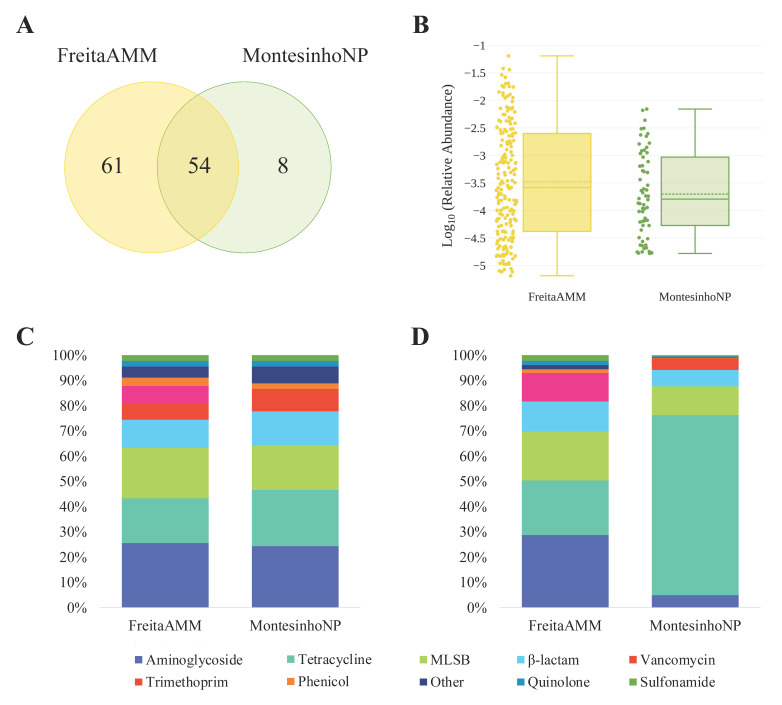
AMR genetic determinants identified on red fox samples based on collection site. (**A**) A Venn diagram illustrating the proportion of shared genes; (**B**) the relative abundance of genes (log_10_); (**C**) the percentage of the variety of ARGs; and (**D**) the abundance by antibiotic class.

**Figure 3 animals-12-02572-f003:**
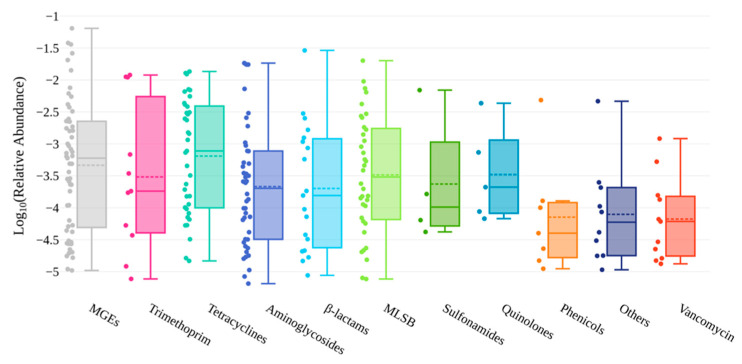
The mean of the relative gene abundances (log_10_ transformed values) for each ARG class and MGEs (including the integrons) detected in the red fox samples under study are represented on the box plot, where error bars represent standard deviation.

**Figure 4 animals-12-02572-f004:**
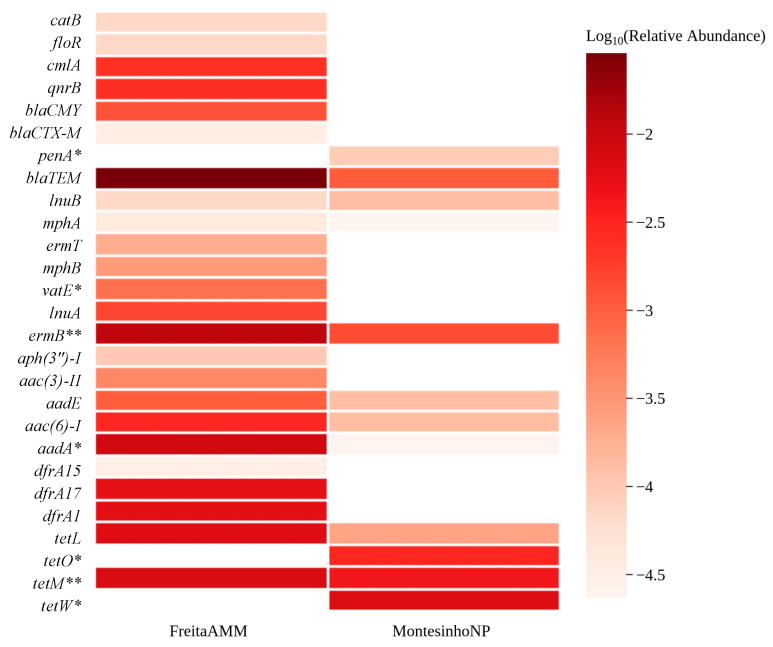
The relative abundance of high-risk ARGs (Rank I and Rank II) identified in this study are shown in the heatmap. * marks ARGs from Rank II, and ** marks ARGs belonging to both ranks.

**Figure 5 animals-12-02572-f005:**
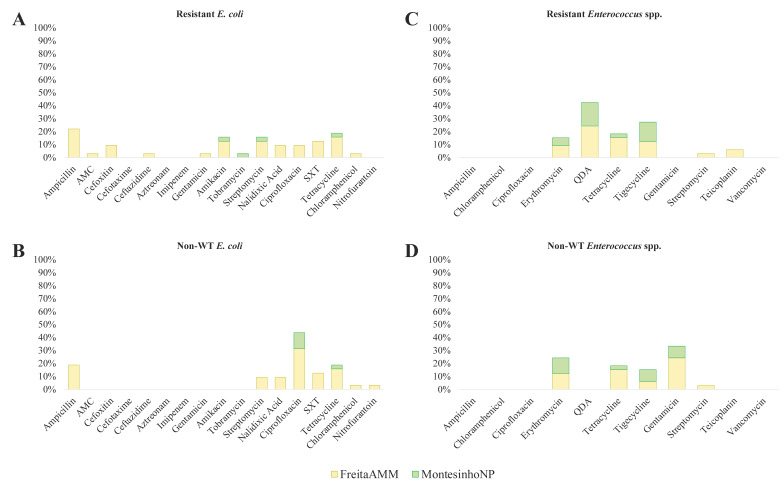
AST results for *E. coli* and *Enterococcus* spp. isolated in the present study, interpreted with clinical breakpoints (resistant strains; (**A**,**C**)) and ECOFFs (non-wildtype strains; (**B**,**D**)). AMC, amoxicillin/clavulanic acid; SXT, trimethoprim–sulfamethoxazole; QDA, quinupristin–dalfopristin.

## Data Availability

The data presented in this study are available on request from the corresponding author.

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
