# Peer review of "Unravelling the Diversity and Abundance of the Red Fox (*Vulpes vulpes*) Faecal Resistome and the Phenotypic Antibiotic Susceptibility of Indicator Bacteria"

_animals, 2022, doi:10.3390/ani12192572_

Round 1

Reviewer 1 Report

This is a nice and interesting article in an area which is relatively under studied. Foxes are a nice way to study what is going on within a wildlife population, and this study very much adds to the literature.

I have only got a few minor comments, which I have detailed below. The vast majority of these are suggested grammatical changes which I have suggested the appropriate change to aid the authors.

Line 21- ‘resistance in red foxes’ may sound better

Line 36- ‘threats to human health’ may sound better

Line 53- ‘transposons and integrons ….’ May sound better

Line 63- ‘wild animals with human activities’ may sound better

Line 86- ‘AMR in red foxes@- may sound better

Line 88- resistome of the red fox – may sound better

Line 99- since the latter are more… (reword)

Line 107- Portugal, bordering Spain … reword

Line 122- is this 22 or 37 as detailed above? If not, why is it 22 and not 37

Line 144- which PCR assay was used, please include details

Line 184- but where other types of samples were analysed ….. reword

Line 194- red foxes from … reword

Line 195- whereas foxes from … reword

Line 198- identified in foxes living in … (reword)

Line 239- whereas no ARG of this … (reword)

Line 258- populations in this study …(Reword)

Line 288- with red foxes reported ….. (reword)

Line 290- perhaps a higher percentage of resistant bacteria?

Line 298 and line 332- here it says Error! Reference source not shown. This may be an editorial issue, but please check

Line 345- Portugal was 179.1 …. (reword)

Line 351- a commonly used quinolone … (reword)

Line 355- foxes confers resistance to … (Reword)

Line 360- that red foxes have the highest … (reword)

Line 366- the AMR dissemination cycle is a bit unclear- please reword

Line 368- with a high level of … (reword)

Author Response

Responses to Reviewer 1 Comments 

The changes made in the manuscript were highlighted in green. 

Reviewer’s comments: 

This is a nice and interesting article in an area which is relatively under studied. Foxes are a nice way to study what is going on within a wildlife population, and this study very much adds to the literature.  

I have only got a few minor comments, which I have detailed below. The vast majority of these are suggested grammatical changes which I have suggested the appropriate change to aid the authors.  

Point 1: Line 21- ‘resistance in red foxes’ may sound better  

Response 1: It was changed. 

Point 2: Line 36- ‘threats to human health’ may sound better  

Response 2: It was changed. 

Point 3: Line 53- ‘transposons and integrons ....’ May sound better  

Response 3: It was changed. 

Point 4: Line 63- ‘wild animals with human activities’ may sound better  

Response 4: It was changed. 

Point 5: Line 86- ‘AMR in red foxes@- may sound better  

Response 5: It was changed. 

Point 6: Line 88- resistome of the red fox – may sound better  

Response 6: It was changed. 

Point 7: Line 99- since the latter are more... (reword)  

Response 7: It was changed. 

Point 8: Line 107- Portugal, bordering Spain ... reword  

Response 8: It was changed. 

Point 9: Line 122- is this 22 or 37 as detailed above? If not, why is it 22 and not 37  

Response 9: From the 37 faecal samples, we selected up to 8 DNAs to set up each pool, giving a n=23. Information was added to the text to clarify it. 

Point 10: Line 144- which PCR assay was used, please include details  

Response 10: The primers and PCR conditions are described in reference 28, co we added this information to the manuscript.  

Point 11: Line 184- but where other types of samples were analysed ..... reword  

Response 11: It was changed. 

Point 12: Line 194- red foxes from ... reword  

Response 12: It was changed. 

Point 13: Line 195- whereas foxes from ... reword  

Response 13: It was changed. 

Point 14: Line 198- identified in foxes living in ... (reword)  

Response 14: It was changed. 

Point 15: Line 239- whereas no ARG of this ... (reword)  

Response 15: It was changed. 

Point 16: Line 258- populations in this study ...(Reword)  

Response 16: It was changed. 

Point 17: Line 288- with red foxes reported ..... (reword)  

Response 17: It was changed. 

Point 18: Line 290- perhaps a higher percentage of resistant bacteria?  

Response 18: It was changed. 

Point 19: Line 298 and line 332- here it says Error! Reference source not shown. This may be an editorial issue, but please check  

Response 19: It was corrected. 

Point 20: Line 345- Portugal was 179.1 .... (reword)  

Response 20: It was changed. 

Point 21: Line 351- a commonly used quinolone ... (reword)  

Response 21: It was changed. 

Point 22: Line 355- foxes confers resistance to ... (Reword)  

Response 22: It was changed. 

Point 23: Line 360- that red foxes have the highest ... (reword)  

Response 23: It was changed. 

Point 24: Line 366- the AMR dissemination cycle is a bit unclear- please reword  

Response 24: It was rewrite (the word “cycle” was removed). 

Point 25: Line 368- with a high level of ... (reword)  

Response 25: It was changed. 

Reviewer 2 Report

The manuscript by Dias et al. is well conducted, with original contribution in the understanding of the characterization of the antimicrobial resistance of the faecal microbiome of the red fox (Vulpes vulpes).

I support its possible publication after appropriate minor modifications as outlined below:

Line 31: „we screened” – I would like to suggest to the authors that please avoid the using of personal mode formulations throughout the manuscript, this is not characteristic for scientific style. Please be carefully with this concern!

Line 40: I suggest the using of „antimicrobials” instead of „antibiotics”

Line 67: „There are still few studies...” – here, please insert references

Line 78, 282, : „[15], [18]” insert a comma before but (!) revise everywhere

Lines 103-118: it would be important for the reader to justify the total number of collected samples. Why exactly 37? In this regard, the authors need to refer to a statistical model related by the total number of foxes living in the monitored region. So, the authors must convince the scientific community that they results are statistically representative and completely supportable by statistical tools.

Line 120: please mention the initial quantity of fecal samples used for DNA extraction

Line 129: “in agreement with Dias et al. [27]” instead of “in agreement with [27].” – please revise this citation issue throughout the manuscript

Line 143: when you mention used reagents (e.g. [Liofilchem, Italy), please uniformly provide the name of producer/company, city and country throughout the manuscript

Line 148: in the “Antibiotic susceptibility testing (AST)” the authors must provide the name of the tested antimicrobials according to their classes. Also, please ensure that there are defined clinical breakpoints in the EUCAST guideline for each of the tested antimicrobials

The recorded susceptibility of the tested strains against vancomycin need to receive particular emphasis meaning its inclusion in discussion

Line 298, 332: “(Figure 5Error! Reference source not found.B, (Figure 5Error! Reference source not found.D) ???

Line 337: the Conclusion section must be substantially shortened, focusing on conclusions derived from the main findings, and presenting the study limitations and future perspectives. In its present form this section seems to be boring for the reader. Thank you for your understanding!

Author Response

Responses to Reviewer 2 Comments 

The changes made in the manuscript were highlighted in turquoise. 

Reviewer’s comments: 

The manuscript by Dias et al. is well conducted, with original contribution in the understanding of the characterization of the antimicrobial resistance of the faecal microbiome of the red fox (Vulpes vulpes).  

I support its possible publication after appropriate minor modifications as outlined below:   

Point 1: Line 31: „we screened” – I would like to suggest to the authors that please avoid the using of personal mode formulations throughout the manuscript, this is not characteristic for scientific style. Please be carefully with this concern!  

Response 1: Personal mode formulations were substituted. 

Point 2: Line 40: I suggest the using of „antimicrobials” instead of „antibiotics”  

Response 2: It was changed. 

Point 3: Line 67: „There are still few studies...” – here, please insert references  

Response 3: References were inserted. 

Point 4: Line 78, 282, : „[15], [18]” insert a comma before but (!) revise everywhere  

Response 4: It was changed. 

Point 5: Lines 103-118: it would be important for the reader to justify the total number of collected samples. Why exactly 37? In this regard, the authors need to refer to a statistical model related by the total number of foxes living in the monitored region. So, the authors must convince the scientific community that they results are statistically representative and completely supportable by statistical tools.  

Response 5: There are a few studies in the North and Central Portugal that studied the factors which influence the presence and density of foxes (Alexandre et al. (2019) and Castro et al. (2022)). In Spain, fox density seems to be linked to food availability, 2.5 ind/km2 and 0.8 foxes ind/km2, from higher to lower, respectively (Yom-Tov et al., 2007). But, a scientific study estimating the fox population size in Portugal is lacking and therefore we were not able to perform what is suggested. Even so, we added the information of a lack of studies estimating fox densities in the main manuscript. 

Point 6: Line 120: please mention the initial quantity of fecal samples used for DNA extraction Response 6: It was added (n=23). 

Point 7: Line 129: “in agreement with Dias et al. [27]” instead of “in agreement with [27].” – please revise this citation issue throughout the manuscript  

Response 7: This was altered. 

Point 8: Line 143: when you mention used reagents (e.g. [Liofilchem, Italy), please uniformly provide the name of producer/company, city and country throughout the manuscript  

Response 8: This was altered. 

Point 9: Line 148: in the “Antibiotic susceptibility testing (AST)” the authors must provide the name of the tested antimicrobials according to their classes. Also, please ensure that there are defined clinical breakpoints in the EUCAST guideline for each of the tested antimicrobials  

Response 9: It was added. 

Point 10: The recorded susceptibility of the tested strains against vancomycin need to receive particular emphasis meaning its inclusion in discussion 

Response 10: It was included.  

Resistance to glycopeptides, important for the treatment of severe human infections, was identified only for 6% of the strains to teicoplanin, whereas no resistance was detected for vancomycin.” 

Point 11: Line 298, 332: “(Figure 5Error! Reference source not found.B,” (Figure 5Error! Reference source not found.D) ???  

Response 11: Now it is correct. 

Point 12: Line 337: the Conclusion section must be substantially shortened, focusing on conclusions derived from the main findings, and presenting the study limitations and future perspectives. In its present form this section seems to be boring for the reader. Thank you for your understanding!  

Response 12: The Conclusions section was changed, as suggested. 

Reviewer 3 Report

This is a very interesting study from the One Health point of view that allows us to analyze the presence of resistance genes in a sentinel species of mammal (the red fox).

However, before publication, some improvements need to be made.

INTRODUCTION

Lines 88-94 and 98-99 correspond to the methodology and should be included in the Materials and Methods section.

MATERIALS AN METHODS

How the samples are collected is not specified. Were they kept in a specific transport medium that would enhance the survival of the bacteria or simply in a sterile container? How long does it take from sample collection to analysis, especially cultures? And how are samples kept during that time (refrigerated or frozen)?

Regarding DNA extraction, the commercial house of the kit used is not specified. Is it the same as in reference 27?

The sample size is quite small and manageable. I understand that the collection of samples in wild fauna is very complex, as you were collecting fresh feces, but why weren't all the samples used to make the pools?

Throughout this section, you talk about the pre-screening of genes and the selection made for the individual analysis of the samples, but the final number of ARGs or MGEs is not indicated (you indicate it in the results, L. 180). Please include this information in Material and Methods.

L. 146. Please change "isolate" to "random strain".

L. 154. I think you forgot to close the parentheses after reference 27.

RESULTS AND DISCUSSION

L. 175 to 193. Please divide the paragraph in two. The second paragraph should start on line 187 with the general results obtained.

Figure 1: Move to line 194 to match the text, please. I suggest including the percentages in the graphs and enlarging the figure for easy reading. Please justify the title of the figure.

Figure 2: please justify the title of the figure.

L. 214-232: this paragraph should be divided into three different ones. The first is up to L. 216, rising the figure 2 to the next line. The second paragraph would start with "The abundance of ARGs...". The third would start with "Overall, the most abundant ARGs...".

L. 298 and 332: Please correct this: "Figure 5Error! Reference source not found.B, Table S4"

CONCLUSIONS

This section must include the conclusions of your study, referring to the data you obtained. Lines 342-352 and 359-365 are considered discussion, not conclusions, so they should be included in the previous section.

Author Response

Responses to Reviewer 3 Comments 

The changes made in the manuscript were highlighted in teal. 

This is a very interesting study from the One Health point of view that allows us to analyze the presence of resistance genes in a sentinel species of mammal (the red fox). However, before publication, some improvements need to be made.  

INTRODUCTION  

Point 1: Lines 88-94 and 98-99 correspond to the methodology and should be included in the Materials and Methods section. 

Response 1: We changed the sentence to avoid confusion with the methodology. Our goal was to give some information about the techniques used. 

MATERIALS AN METHODS  

Point 2: How the samples are collected is not specified. Were they kept in a specific transport medium that would enhance the survival of the bacteria or simply in a sterile container? How long does it take from sample collection to analysis, especially cultures? And how are samples kept during that time (refrigerated or frozen)? 

Response 2: The information was added. 

Point 3: Regarding DNA extraction, the commercial house of the kit used is not specified. Is it the same as in reference 27?  

Response 3: We added that information. 

Point 4: The sample size is quite small and manageable. I understand that the collection of samples in wild fauna is very complex, as you were collecting fresh feces, but why weren't all the samples used to make the pools?  

Response 4: In several studies, sample pooling has proven to be highly effective when using a sufficiently sensitive technique such as qPCR. We had a limited budget to the qPCR analysis, so we had to select a few representative samples. Since our goal was to generally gain insights in the general resistome of red fox, we adopted this strategy and selected samples from different years of collection, to get diversity. To clarify this, we added information to the manuscript and we also included the need to pool samples as a limitation of the study, in the conclusions section. 

Point 5: Throughout this section, you talk about the pre-screening of genes and the selection made for the individual analysis of the samples, but the final number of ARGs or MGEs is not indicated (you indicate it in the results, L. 180). Please include this information in Material and Methods.  

Response 5: The information were included in the Material and Methods section. 

Point 6: L. 146. Please change "isolate" to "random strain".  

Response 6: It was changed. 

Point 7: L. 154. I think you forgot to close the parentheses after reference 27.  

RESULTS AND DISCUSSION  

Response 7: It was verified. 

Point 8: L. 175 to 193. Please divide the paragraph in two. The second paragraph should start on line 187 with the general results obtained.  

Response 8: It was changed. 

Point 9: Figure 1: Move to line 194 to match the text, please. I suggest including the percentages in the graphs and enlarging the figure for easy reading. Please justify the title of the figure.  

Response 9: The figure was moved, enlarged, and the percentages were added. The text was justified. 

Point 10: Figure 2: please justify the title of the figure.  

Response 10: It was justified. 

Point 11: L. 214-232: this paragraph should be divided into three different ones. The first is up to L. 216, rising the figure 2 to the next line. The second paragraph would start with "The abundance of ARGs...". The third would start with "Overall, the most abundant ARGs...".  

Response 11: The block was divided into three paragraphs, and Fig. 3 was moved, as suggested. 

Point 12: L. 298 and 332: Please correct this: "Figure 5Error! Reference source not found.B, Table S4"  

Response 12: It was done. 

CONCLUSIONS  

Point 13: This section must include the conclusions of your study, referring to the data you obtained. Lines 342-352 and 359-365 are considered discussion, not conclusions, so they should be included in the previous section.  

Response 13: It was changed, as suggested.